



# The long-term impact of transgressing planetary boundaries on biophysical atmosphere-land interactions

Markus Drüke[1], Wolfgang Lucht[1,2], Werner von Bloh[1], Stefan Petri[1], Boris Sakschewski[1], Arne Tobian[3,1], Sina Loriani[1], Sibyll Schaphoff[1], Georg Feulner[1], and Kirsten Thonicke[1]

[1]Potsdam Institute for Climate Impact Research, Member of the Leibniz Association, Telegraphenberg A31, 14473 Potsdam, Germany
[2]Department of Geography, Humboldt Universität zu Berlin, Unter den Linden 6, 10099 Berlin, Germany
[3]Stockholm Resilience Centre, Stockholm University, Kräftriket 2B, 10691 Stockholm, Sweden

**Correspondence:** Markus Drüke (drueke@pik-potsdam.de)

**Abstract.** Human activities have had a significant impact on Earth's systems and processes, leading to a transition of Earth's state from the relatively stable Holocene epoch to the Anthropocene. The planetary boundaries framework characterizes major risks of destabilization, particularly in the core dimensions of climate and biosphere change. Land system change, including deforestation and urbanization, alters ecosystems and impacts the water and energy cycle between land surface and atmosphere,

5   while climate change can disrupt the balance of ecosystems and impact vegetation composition and soil carbon pools. These drivers also interact with each other, further exacerbating their impacts. Earth system models have been used recently to illustrate the risks and interacting effects of transgressing selected planetary boundaries, but a detailed analysis is still missing. Here, we study the impacts of long-term transgressions of the climate and land system change boundaries on the Earth system using an Earth system model with an incorporated detailed dynamic vegetation model. In our centennial-scale simulation

10   analysis, we find that transgressing the land system change boundary results in increases in global temperatures and aridity. Furthermore, this transgression is associated with a substantial loss of vegetation carbon, exceeding 200 PgC, in contrast to conditions considered safe. Concurrently, the influence of climate change becomes evident as temperatures surge by 2.7-3.1 °C depending on the region. Notably, carbon dynamics are most profoundly affected within the large carbon reservoirs of the boreal permafrost areas, where carbon emissions peak at 150 PgC. While a restoration scenario to reduce human pressure

15   to meet the planetary boundaries of climate change and land system change proves beneficial for carbon pools and global mean temperature, a transgression of these boundaries could lead to profoundly negative effects on the Earth system and the terrestrial biosphere. Our results suggest that respecting both boundaries is essential for safeguarding Holocene-like planetary conditions that characterize a resilient Earth system and are in accordance with the goals of the Paris Climate Agreement.



## 1 Introduction

Human activities have had a significant impact on important processes and interactions in the Earth system, increasingly shifting the planet away from its historically known state. The influence of human actions on the Earth system is now approaching the same order of magnitude as that of longer-term geological processes. This leads to Earth's transition from the relatively stable Holocene epoch to the newly termed Anthropocene (Steffen et al., 2018). In particular, land use and the emissions caused by fossil fuel use introduce significant perturbations to the biosphere, which is a central pillar of stability of the Earth system as a whole. Humanity is currently at a critical juncture, where there is an opportunity to limit change to Earth's climate to a magnitude that is likely still manageable, while insufficient action would lead to Earth leaving the historically known "safe operating space" for humanity (within society's adaptive capacities) and force future generations to live in an unstable, warmer climate and a fundamentally transformed biosphere (Rockström et al., 2009; Steffen et al., 2018; Díaz et al., 2019).

To better understand the limits to Earth system change that reasonably characterize a safe operating space for humanity, i.e. the planetary boundaries, it is crucial to study the relevant Earth system processes and their interactions. These processes can be amplified through feedback cycles, where the change in one quantity (such as increased climatic warming) leads to a change in another (such as global forest cover), which then in turn again impacts the first quantity. These feedback mechanisms can either amplify or dampen the effect of climate forcing or land use. Many of these feedback cycles are connected to the functioning of the terrestrial biosphere, which is closely linked to the atmosphere through the carbon, water, and energy cycles via processes such as vegetation growth, evapotranspiration, albedo, and roughness length. By identifying key Earth system processes and establishing planetary boundaries for each, the planetary boundary framework (Rockström et al., 2009) helps to identify the risks of destabilization of the biosphere due to human activities.

Land system change, which includes deforestation and urbanization, is a key process that has been defined as one of the nine planetary boundaries. It is a major driver of environmental change by impacting the functioning of ecosystems and contributing to the loss of biodiversity (Díaz et al., 2019). The transgression of this boundary can also have a range of other impacts, such as altering the carbon cycle and contributing to climate change (IPCC, 2023). The planetary boundary for land system change is defined through the observable proxy given by the extent of remaining forest cover (50%, 85%, and 85%) of the three major forest biomes – temperate, tropical, and boreal, respectively (Steffen et al., 2015).

Climate change can disrupt the balance of ecosystems and result in shifting global temperature and precipitation patterns. It is also expected to lead to increasing frequency and severity of extreme events and altered availability of water resources (IPCC, 2023). The planetary boundary framework sets the lower boundary of the uncertainty range for the transgression of atmospheric $CO_2$ concentration at 350 ppm, a value which was reached in 1988 (Rockström et al., 2009; Steffen et al., 2015).

Both of these first-order drivers, land use change and climate change, can simultaneously impact carbon dynamics and global climate, further exacerbating the overall impact (Foley et al., 2011). For instance, the global climate directly influences energy and water cycles, while concurrently, alterations in transpiration dynamics and albedo resulting from land use changes contribute to additional shifts in vital systems.

In the past, the planetary boundary framework relied heavily on expert knowledge and observation, as well as on the analysis




of data and evidence from a variety of sources, including satellite observations, field measurements, and simplified models
(Rockström et al., 2009; Steffen et al., 2015). However, there is still a lack of systematic exploration and study to examine
the combined impact of several planetary boundaries using comprehensive Earth system models. Recently, Richardson et al.
(2023) published the third major update of the planetary boundary framework, applying for the first time results from an Earth
system model to study the interaction of a limited number of planetary boundaries.

In this study, we analyze the combined impact of crossing the planetary boundaries for climate change (CC) and land system
change (LSC), by applying the Potsdam Earth model (POEM, Drüke et al., 2021b), which includes a process-based repre-
sentation of the biosphere's response to climate change. In particular POEM's advanced vegetation model component LPJmL
(Schaphoff et al., 2018b) is able to realistically account for important processes connected to plant productivity, phenology, and
mortality. We performed simulations using the POEM model to examine the impact of different climate forcings (atmospheric
$CO_2$ levels of 350, 450, and 550 ppm) and land-use patterns (lower, upper and transgressed, see methods) on carbon stocks
and global climate, following the planetary boundary framework (Rockström et al., 2009; Steffen et al., 2015). We separately
analyzed the short-term (1980-2100) and long-term (2100-2770) changes in carbon stocks, temperature, and precipitation to
understand the potential biological and physical interactions under different forcing conditions.

While the IPCC focuses its work strongly on transient scenarios until 2100 (RCP scenarios, Myhre et al., 2013; IPCC, 2023),
we here aim to contribute a model-based analysis motivated by a somewhat different intent, namely to study the long-term
implications of various levels of commitment stemming from Earth system change of a fixed magnitude. The interconnected
effects of a prescribed anthropogenic climate and land system alterations strongly impact energy and water fluxes between
the terrestrial biosphere and atmosphere, which has severe long-term repercussions for the Earth system. A structural change
of vegetation cover through, e.g., deforestation, impacts albedo, roughness length, surface roughness, evapotranspiration, and
surface temperatures. A change in atmospheric forcing, on the other hand, indirectly impacts natural and managed land, by
increasing temperatures, changing precipitation patterns, and a higher frequency of extreme events.

This present study documents and explores the prospective transformations of the land carbon cycle and global climate follow-
ing from changes in climate forcing and land system change. Building upon the foundation laid by Richardson et al. (2023), this
paper aims for a more detailed analysis, delving in more detail into the feedback mechanisms between the biosphere and the
atmosphere. While Richardson et al. (2023) predominantly scrutinized overarching global metrics such as carbon accumulation
and emissions, our study focuses on the elucidation of spatially and temporal explicit outcomes that arise upon transgressing
the planetary thresholds of climate and land system change.

## 2 Methods

### 2.1 Model description

POEM (in the configuration of CM2Mc-LPJmL, Drüke et al., 2021b) is an Earth system model that combines the atmosphere
and ocean model CM2Mc (Galbraith et al., 2011), which has a coarse spatial resolution and hence is relatively fast, with
the dynamic global vegetation model (DGVM) LPJmL version 5 (LPJmL5, Schaphoff et al., 2018b; Von Bloh et al., 2018).



CM2Mc is based on the Climate Model CM2 (Milly and Shmakin, 2002) and includes the Modular Ocean Model 5 (MOM5) and the global atmosphere and land model AM2-LM2 or AM2-LM (Anderson et al., 2004) with static vegetation. In POEM, the land component LM/LM2 has been replaced by the DGVM LPJmL5, while AM2 and MOM5 remain dynamically coupled
to the model. The different model components are connected using the Flexible Modeling System (FMS), developed by GFDL. LPJmL5 (Lund-Potsdam-Jena managed Land, Schaphoff et al., 2018b) is a state-of-the-art DGVM, simulating the water and carbon fluxes, carbon stocks, and global surface energy balance for natural and managed land. The model takes into account bioclimatic limits and the effects of heat, productivity, and fire on plant mortality to determine the establishment, growth, competition, and mortality of different plant functional types (PFTs) in natural vegetation and crop functional types (CFTs)
on managed land. It is driven by climate and soil data. Since its original publication (Sitch et al., 2003), LPJmL5 has been enhanced to include a water balance (Gerten et al., 2004), agriculture (Bondeau et al., 2007), wildfire in natural vegetation (Thonicke et al., 2010; Drüke et al., 2019), and the impact of multiple climate drivers on phenology (Forkel et al., 2014, 2019), which was described in Schaphoff et al. (2018b) and extensively evaluated in Schaphoff et al. (2018a). LPJmL5 is described in Von Bloh et al. (2018).

These processes within LPJmL5 significantly influence the location, timing, and magnitude of atmospheric water fluxes and precipitation through plant evapotranspiration, surface temperature, and simulated canopy humidity. The fully coupled energy and water cycle enables the investigation of the impact of biophysical atmosphere-biosphere feedbacks on global climate trajectories and the quantification of the impacts of deforestation or afforestation scenarios (Drüke et al., 2023).

In contrast to the previously published configuration of POEM, CM2Mc-LPJmL (Drüke et al., 2021b), this updated version
incorporates an adjustment where we have set the net water influx from rivers into the ocean to zero. This modification serves to mitigate model-induced fluctuations in sea level, while allowing that sea level changes can still occur as a result of processes like sea ice melt and thermal expansion.

## 2.2 Development of LSC scenarios

Scenario-specific land-use datasets, representing different levels of the transgression of the land-use change planetary boundary,
were compiled using the following algorithm:

The status of the land system change boundary is determined by comparing the current extent of each of the three major forest biomes (tropical, temperate, and boreal) on each continent (see Table S1) to their potential natural vegetation extent (vegetation data after spinup before the transient phase, without human land use). The tropical, temperate, and boreal climate zones used here are based on the Koeppen and Geiger scheme (S1-S3, Kottek et al., 2006) and are identified using a biome
classification based on Ostberg et al. (2013). Depending on the scenario, the algorithm either allows for the expansion of the biome by reducing land cover (afforestation) or replacing natural vegetation with agricultural land cover (deforestation). For afforestation, the local land-use fraction is multiplied by a reduction factor to meet the scenario-specific extent. The algorithm for deforestation involves two steps: intensification and expansion. Intensification occurrs in cells where anthropogenic land use already exists, increasing the preexisting land use until the scenario target is reached or the cell is completely depleted
of the forest biome. Expansion affected cells that are not currently under anthropogenic land use and involve populating each




unpopulated cell within the target biome with the land-use mix of neighboring cells, if applicable. In the next iteration, the freshly populated cells can again be intensified or have land-use spread until the scenario condition is met.

## 2.3 Model protocol

The model experiments of this paper are consistent with Drüke et al. (2021b). A 5000-year stand-alone spin-up of the LPJmL5 model is performed first. This is followed by a fully coupled spin-up under pre-industrial conditions for 1500 model years, ensuring a consistent equilibrium between the long-term soil carbon pool, vegetation, ocean, and climate. The historic period is then simulated from 1700 to 2018, using historic land use data from 1700 (Fader et al., 2010) and historic concentrations of atmospheric greenhouse gases, solar radiation, ozone concentrations, and aerosols from 1860, which were kept at pre-industrial

conditions beforehand (see Drüke et al., 2021b). From 2004 on, only greenhouse gas forcing is changed, while aerosols, solar radiation, and ozone are set to their corresponding 2003 values.

From 2019 on, we switch from the historic forcing to scenarios and perform our model runs following a setting where the planetary boundaries for climate change (CC) and land system change (LSC) are set to pre-determined scenario values for another 800 years. The long temporal span was chosen to allow insights into the evolution of key climatic variables and their

impact on the terrestrial carbon balance. The study examines six different scenarios, ranging from low to high levels of land use change and climate change, respectively. Scenario values for atmospheric $CO_2$ concentration are set to 350, 450, and 550 ppm, while values for land system change (pattern of the remaining forest in tropical, temperate, and boreal forests) are generated by an algorithm, described in Sect. 2.2. In the scenario names, "Lower" and "Upper" refer to the zone of increasing risk of the planetary boundaries for CC and LSC, with the "Lower" end representing the boundary defining the "safe operating space" and

"Upper" representing the transition from risky to dangerous. "Transgressed" refers to a forcing that exceeds the upper end of the zone of increasing risk (for an overview see Fig. 1). The lower boundary of the CC boundary was transgressed in 1988. Consequently, we incorporated a scenario in which conditions remained constant from that year onward and compared the long-term effects on climate and carbon pools to the conditions specifically established in 1988 (for land use and $CO_2$ time series for all scenarios see Figs. S4 and S5). The full simulation protocol follows the one in the latest update of the planetary

boundary framework (Richardson et al., 2023).

1. Lower CC and 1989 values of LSC: From 1989, atmospheric $CO_2$ concentrations are kept constant at 350 ppm and the land-use pattern from 1989 is maintained for 800 years.

2. Upper CC and lower LSC: From 2019, the percentage of remaining natural vegetation is increased to 85% in the tropical, 50% in temperate and 85% in the boreal zone following a linear trajectory, and the atmospheric $CO_2$ concentrations are

increased by 1% per year from 2019 levels to reach 450 ppm. From 2029, the land-use pattern is kept constant, and atmospheric $CO_2$ concentrations are maintained at 450 ppm for 800 years.

3. Upper CC and upper LSC: From 2019, the percentage of remaining natural vegetation is decreased to 60% in tropical, 30% in temperate, and 60% in boreal regions, following a linear trajectory and atmospheric $CO_2$ concentrations are



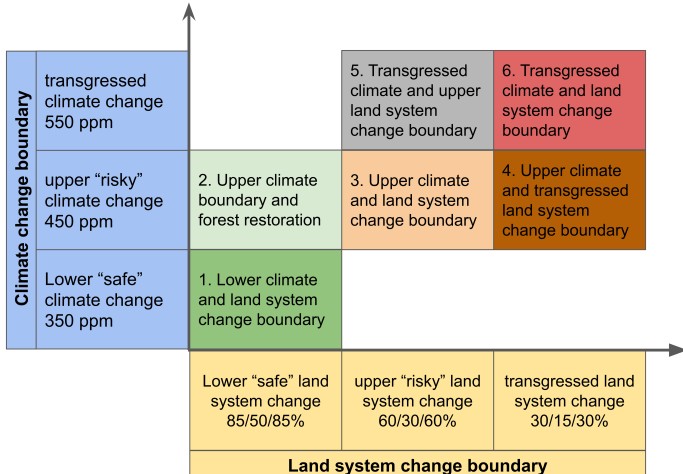

**Figure 1.** Overview of simulation experiments for the effect of the climate change (CC) and land system change (LSC) planetary boundaries. The percent of different states of LSC denotes natural land remaining in the tropical, temperate or boreal zone, respectively.

increased by 1% per year from 2019 levels to reach 450 ppm. From 2029, the land-use pattern is kept constant, and atmospheric $CO_2$ concentrations are maintained at 450 ppm for 800 years.

4. Upper CC and transgressed LSC: From 2019, the percentage of remaining natural vegetation is decreased to 40% in tropical, 20% in temperate, and 40% in boreal regions, following a linear trajectory and atmospheric $CO_2$ concentrations are increased by 1% per year from 2019 levels to reach 450 ppm. From 2029, the land-use pattern is kept constant, and atmospheric $CO_2$ concentrations are maintained at 450 ppm for 800 years.

5. Transgressed CC and upper LSC: From 2019, the percentage of remaining natural vegetation is decreased to 60% in tropical, 30% in temperate, and 60% in boreal regions, following a linear trajectory and atmospheric $CO_2$ concentrations are increased by 1% per year from 2019 levels to reach 550 ppm. From 2052, the land-use pattern is kept constant, and atmospheric $CO_2$ concentrations are maintained at 550 ppm for 800 years.

6. Transgressed CC and LSC: From 2019, the percentage of remaining natural vegetation is decreased to 40% in tropical, 20% in temperate, and 40% in boreal regions, following a linear trajectory and atmospheric $CO_2$ concentrations are increased by 1% per year from 2019 levels to reach 550 ppm. From 2052, the land-use pattern is kept constant, and atmospheric $CO_2$ concentrations are maintained at 550 ppm for 800 years.

In this study, the term "land system change" (LSC) is used interchangeably with "land-use change". The LSC boundary is defined by the modification of land use patterns and the associated social and ecological systems that result from human decision-making at various scales. However, our current modeling capabilities only allow for a reliable representation of land-use change. Therefore, land-use change is selected as the representative proxy for the LSC boundary.



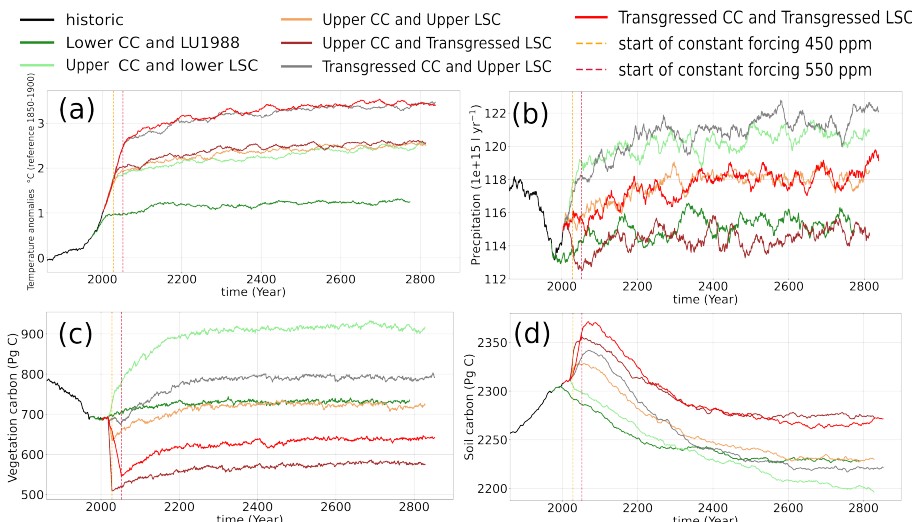

**Figure 2.** Global time series of (a) 30-year running mean of land surface temperature, (b) 30-year running mean of land precipitation in, (c) vegetation carbon and (d) soil carbon for the different simulation experiments 2020-2770 with the historic period until 2019 shown in black. The vertical dashed lines mark the year from which the atmospheric $CO_2$ concentration was held constant.

## 3 Results and discussion

This study focuses on an analysis for six distinct simulation scenarios for 750-800 years, each formulated to represent varying states of CC and LSC planetary boundary transgressions. Notably, during the initial decade of experimentation involving the
450 ppm scenarios and the first four decades involving the 550 ppm scenarios (called thereafter the transient phase), we forced the gradual transition from 2019-level conditions to the designated final state of the prescribed boundaries, encompassing atmospheric carbon concentration and the extent of land use alteration. In our results, we focus on the long-term impact of the planetary boundaries for climate change and land system change on important Earth system variables as temperature, precipitation, and carbon stored in vegetation and soil.

### 3.1 Temperature

Globally and across all scenarios the global mean land temperature development follows atmospheric $CO_2$ forcing as a sharp increase in the transient phase, while further temperature evolution shows a delay for almost the complete study period to approach a new equilibrium in a warmer world (Fig. 2a), where notable alterations in global temperature are observed (see Fig. 3). In all experiments, even at a constant 350 ppm atmospheric $CO_2$ concentration, temperatures increase throughout most
areas globally compared to 1988 values (0.61–0.73 °C at lower CC/ lower LSC to 2.7–3.1 °C at transgressed CC/transgressed LSC, see Fig. 4). The conjunction of these findings with Fig. 2a proves a significant lag of several centuries in the adjustment of global temperature to novel $CO_2$ levels, highlighting the marked distinction between transient climate change and the long-term climate equilibrium. This lag in temperature increase can mostly be attributed to the buffering effect of the oceans which have



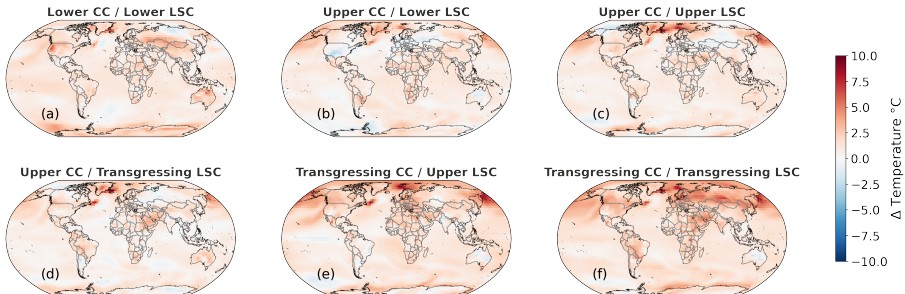

**Figure 3.** Impact of different CC and LSC planetary boundary states on global surface temperature pattern as a mean over 30 years of the experiment (2740-2770) compared to 1988.

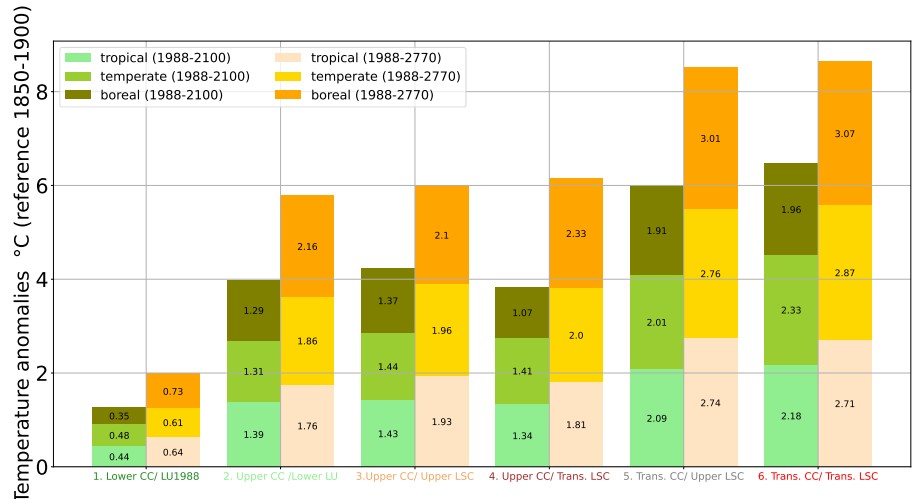

**Figure 4.** Impact of different CC and LSC planetary boundary states on average surface land temperature for the tropical (light green), temperate (forest green) and boreal zone (olive green) from 1988-2100 (short-term) and for the tropical (light red), temperate (yellow) and boreal zone (orange) from 1988-2770 (long-term). The individual temperature difference for each climatic zone is denoted on the corresponding part of the stacked bars. The vertical axis illustrates the cumulative temperature increase for the three zones, enhancing clarity. Notably, this does not represent the overall global land temperature.

a large heat capacity and store more than 90% of the heat associated with observed greenhouse-gas-attributed global warming
(Durack et al., 2014). All three major climate zones follow this trend as for instance in the Upper CC/Upper LSC scenario the short-term temperature changes from 1988-2100 are 1.4 °C for tropical, temperate and boreal areas and 1.9/2.0/2.1 °C for the long-term changes from 1988-2770, therefore almost 30% of the total increase in temperatures develops after 2100 with constant conditions from 2030 (Fig. 4). Boreal regions, however, show an accelerated warming compared to other areas. For instance, in the Transgressed CC/Transgressed LSC scenario, the boreal region warms by 3.1 °C on average in the modeling
period, compared to 2.7 °C in the tropical zone and 2.9 °C in the temperate zone (Fig. 4). Here, the decrease of snow cover





due to increasing temperatures leads to a lower albedo, influencing Earth's reflectivity. Consequently, this triggers heightened absorption of solar radiation and a corresponding temperature increase (Serreze and Barry, 2011). Therefore, the increase in Arctic temperatures is larger compared to the global average, a phenomenon called the Arctic amplification (Screen and Simmonds, 2010).

Global land temperatures are strongly related to alterations in land use change. In regions where substantial land use modifications have occurred, such as in tropical zones, the rate of temperature increase outpaces that of regions with relatively stable land utilization (e.g. Transgressed CC/Upper LSC vs. Transgressed CC/Transgressed LSC, Fig. 3e and f). This effect is mainly due to modified transpiration dynamics and the roughness length effect: Deforestation reduces transpirational water flux, as crops release less water into the atmosphere (Gkatsopoulos, 2017). Consequently, this curtails atmospheric water flux, resulting

in diminished humidity, altered precipitation patterns, and a cooling reduction facilitated by latent heat exchange. Additionally, a decrease in natural vegetation cover decreases the roughness length of the land surface, further increasing temperatures and accentuating the interplay between land cover and climatic conditions (Hoffmann and Jackson, 2000). On the other hand, crops and grassland have an elevated albedo profile compared to closed forest canopies (Unger, 2014). This variance in surface reflectivity results in a discernible shift in the energy cycles, changing them toward a diminished surface temperature.

Collectively, the sum of these biophysical feedback mechanisms leads to a climate characterized by increased temperatures across extensive geographic extents. This holds particularly in tropical latitudes, where increased sensible heat fluxes surpass the potential cooling by elevated surface albedo.

In contrast, instances of augmented natural land, as observed in scenarios characterized by lower LSC, conceivably contribute to localized cooling effects. This is particularly evident in select regions (e.g., parts of Europe within the Upper CC/Lower

LSC scenarios, Fig. 3b). This local cooling in agricultural areas is, however, overshadowed by the overarching warming trend. Minor instances of cooling over oceanic regions can be explained by model uncertainties and stochastic variability. In global average numbers, warming and cooling due to land use change is not very dominant due to only a small fraction of the Earth's surface being managed (Fig. 2a). Therefore, the impact of land use change on global temperature is (in contrast to localized effects) relatively small and in some cases within the range of natural climate variability.

### 3.2 Precipitation

A consistent trend of increased global land precipitation across all scenarios is observed during the study period spanning from 1988 to 2770 (Fig. 2b), while most of this increase happens before 2100 (Fig. 6). This rise in precipitation exhibits a proportional relationship with warming, a correlation attributed to the fundamental Clausius-Clapeyron principle that warmer air can retain more moisture which translates into more moisture transport and precipitation. In particular regions such as South-

East Asia, North America, and Eurasia, show higher annual precipitation, especially in the Upper CC/Lower LSC and Upper CC/Upper LSC experiments (Fig. 5). Notably, the tropical belt stands out with the largest absolute precipitation changes over the study period, due to being the planet's precipitation-richest zone. These disparities in the tropical region encompass both amplifications and reductions in precipitation, consequently yielding a relatively modest spatial sum of precipitation changes in this zone (Fig. 6). In contrast, the temperate zone stands out with the largest cumulative surplus in precipitation from 1988





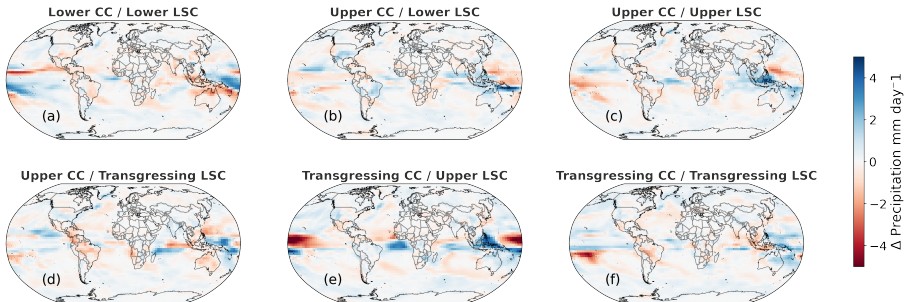

**Figure 5.** Impact of different CC and LSC planetary boundary states on global precipitation pattern as a mean over the 30 years of the experiments (2740-2770) compared to 1988.

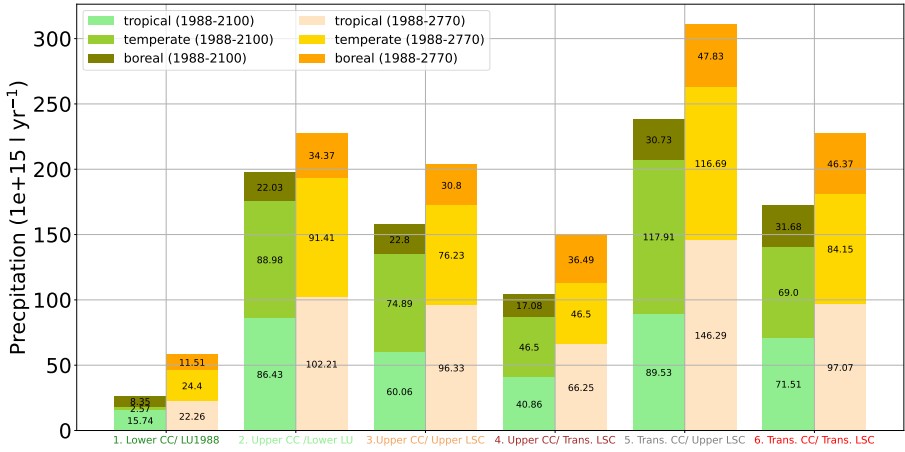

**Figure 6.** Impact of different CC and LSC planetary boundary states on land precipitation for the tropical (light green), temperate (forest green), and boreal zone (olive green) from 1988-2100 (short-term) and for the tropical (light red), temperate (yellow) and boreal zone (orange) from 1988-2770 (long-term). The individual precipitation difference for each climatic zone is denoted on the corresponding part of the stacked bars. Positive numbers denote an increase in precipitation over the corresponding time periods.

to 2770 in all scenarios.

Land use change exerts discernible influences on precipitation trends, steering them towards drier conditions. Although the overall global difference in precipitation relative to 1988 levels remains positive, scenarios characterized by substantial LSC witness a shift toward drier conditions compared to scenarios involving a lower degree of LSC, even with an equivalent level of CC transgression. Illustratively, the transition from Upper CC/Upper LSC to Upper CC/Transgressed LSC leads to diminishing

precipitation differences from 1988 to 2770 in the temperate zone, declining from 76 to $47 \times 10^{15}$ l yr$^{-1}$ (Fig. 6). A contrasting trend emerges for scenarios transitioning from Upper CC/Upper LSC to Transgressed CC/Upper LSC, wherein precipitation deviations surge from 76 to $121 \times 10^{15}$ l yr$^{-1}$ in the temperate zone. Experiments with a larger amount of LSC transgression have slightly less precipitation due to limited evapotranspiration from crops compared to natural forests.



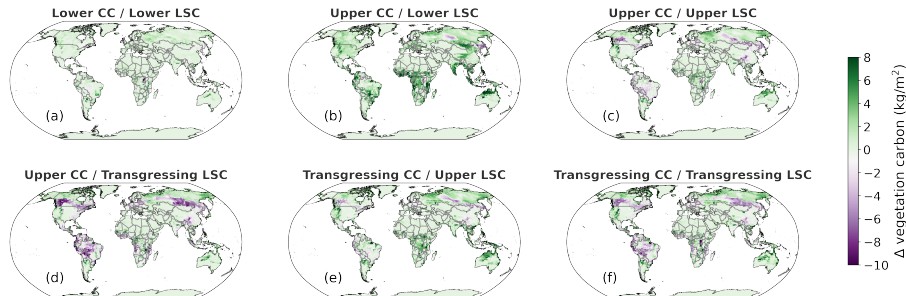

**Figure 7.** Impact of different CC and LSC planetary boundary states on global vegetation carbon pattern as an average over 30 years of the experiment (2740-2770) compared to 1988.

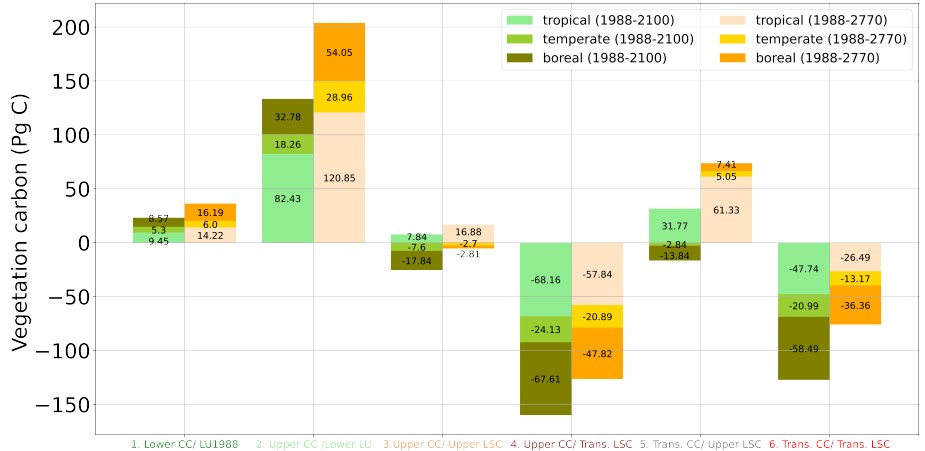

**Figure 8.** Impact of different CC and LSC planetary boundary states on the vegetation carbon pool for the tropical (light green), temperate (forest green), and boreal zone (olive green) from 1988-2100 (short-term) and for the tropical (light red), temperate (yellow) and boreal zone (orange) from 1988-2770 (long-term). The individual carbon difference for each climatic zone is denoted on the corresponding part of the stacked bars. Positive numbers denote a net increase in vegetation carbon over the corresponding time periods and negative numbers a net emission to the atmosphere.

## 3.3 Vegetation carbon

Global projections of vegetation carbon exhibit distinctive patterns, marked by a temporary decline during the initial transient phase in scenarios characterized by pronounced LSC effects. Subsequently, these projections demonstrate a prolonged upward trend, culminating in scenario-specific quasi-equilibrium states emerging around 400-600 years into the simulation period (Fig. 2c).

The transient phase of our study primarily reveals the dominant influence of land system change (LSC) on short-term dynamics.
Particularly noteworthy is the swift decline in vegetation carbon pools resulting from deforestation in scenarios characterized





by high levels of LSC transgression (Figs. 2c and 8). For instance, in the Upper CC/Upper LSC vegetation carbon pools slightly shift by $+7.8/-7.6/-18$ PgC (tropical/temperate/boreal) from 1988-2100. In contrast, in the Upper CC/ Transgressed LSC scenario the pool significantly changes by $-68$ PgC in tropical, $-24$ PgC in temperate, and $-68$ PgC in boreal regions. Moreover, the impact of LSC extends to the geographical distribution of vegetation carbon. Regions with a higher prevalence of agri-

cultural lands, such as Brazil and Europe, exhibit a reduction in vegetation carbon content in scenarios featuring heightened LSC levels (Fig. 7). This spatial pattern contributes significantly to the reduction of both short-term and long-term average vegetation carbon pools across tropical, temperate, and boreal zones. For example, in the Transgressed CC/Upper LSC scenario, tropical vegetation carbon experiences a robust increase of 61 PgC between 1988 and 2770. In contrast, the Transgressed CC/Transgressed LSC scenario sees a notable decline of 26 PgC within the same time frame.

The role of atmospheric $CO_2$ levels is evident, with larger transgressions across the CC boundary corresponding to increased vegetation carbon values (Figs. 7 and 8). For example, under the Upper CC/Upper LSC scenario, where atmospheric $CO_2$ concentrations are at 450 ppm, the tropical vegetation carbon pool increases by 17 PgC over the period from 1988 to 2770. At 550 ppm, within the Transgressed CC/Upper LSC scenario, vegetation carbon increases more substantially by 61 PgC. This global upsurge in the vegetation carbon pool is connected to the carbon dioxide fertilization effect, arising from elevated

atmospheric $CO_2$ concentrations, as well as longer growing seasons due to warming. Full adjustment to the new atmospheric $CO_2$ concentration is a long-term process, taking several hundred years. In the northern boreal zone, climate change fosters augmented vegetation carbon content further, as conditions become more favorable for the establishment of larger plant species and trees within regions initially limited by cold temperatures (Fig. 7f). While assessing the ramifications of climate change, our study reveals a clear dominance of carbon fertilization effects over increasing heat stress (e.g. decreased productivity) and

heat-related mortality (e.g. more frequent and intense wildfires) in influencing global vegetation carbon dynamics.
The temporal dynamics of the impact of CC and LSC vary distinctly. The response time of LSC is relatively swift, exerting a substantial influence on short-term carbon pools, whereas climate change yields a prolonged effect, continuously enhancing vegetation carbon content over an extended time span. Across all scenarios, by 2770 vegetation carbon surpasses 2100 levels, largely due to the delayed impact of carbon fertilization, even under constant atmospheric $CO_2$ levels (Fig. 2c and 8). For

instance, within the Upper CC/Lower LSC scenario, tropical carbon accumulation between 1988 and 2100 amounts to 82 PgC, which increased to 121 PgC between 1988 and 2770, signifying an approximate 50% increase (Fig. 8).

## 3.4   Soil carbon

In contrast to vegetation carbon, global soil carbon pools exhibit a peak shortly after the transient phase and show subsequently

a prolonged downward trend (Fig. 2d). While the peak is largest for Transgressed LSC scenarios, it diminishes for a lower LSC transgression and vanishes for the 1988LU scenario. Swift conversions of vegetation carbon to litter, followed by integration into the soil carbon pool during land use change in the transient phase, predominantly lead to an increase in soil carbon pools in the first decades of the scenarios.
However, between 2100 and 2770, the effect of increasing temperatures exhibits a more detrimental influence on soil carbon





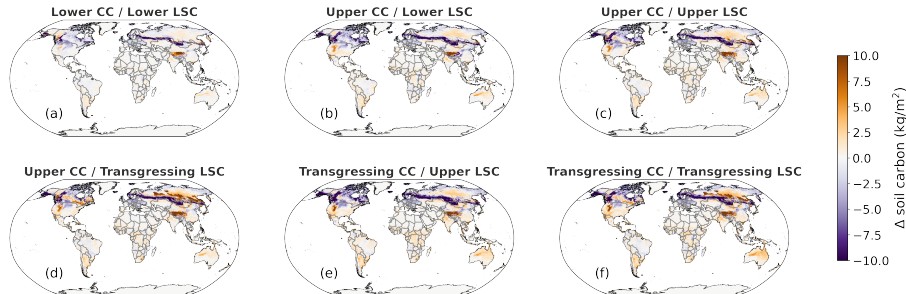

**Figure 9.** Impact of different CC and LSC planetary boundary states on global soil carbon pattern as an average over the last 30 years of the experiment (2740-2770) compared to 1988.

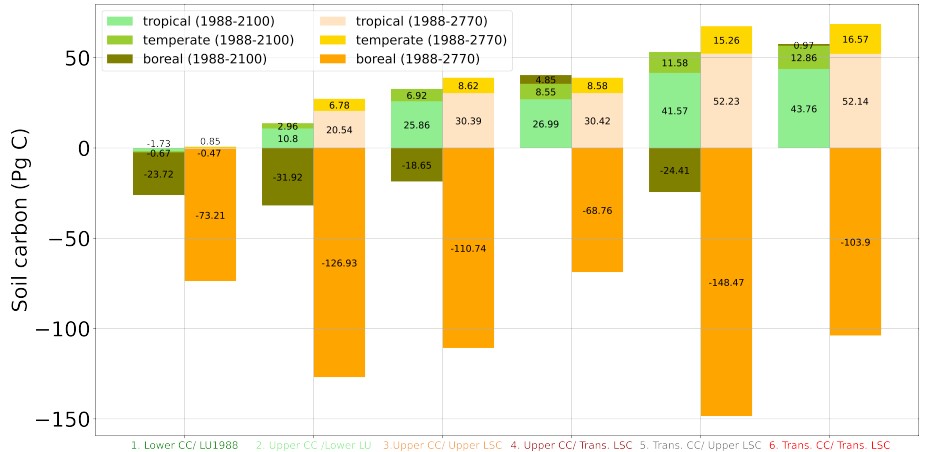

**Figure 10.** Impact of different CC and LSC planetary boundary states on the soil carbon pool for the tropical (light green), temperate (forest green), and boreal zone (olive green) from 1988-2100 (short-term) and for the tropical (light red), temperate (yellow) and boreal zone (orange) from 1988-2770 (long-term). The individual carbon difference for each climatic zone is denoted on the corresponding part of the stacked bars. Positive numbers denote a net increase in soil carbon over the corresponding time periods and negative numbers a net emission to the atmosphere.

dynamics, especially in the boreal zone (Figs. 9 and 10). For instance, in the Transgressed CC/Upper LSC scenario, between 1988 and 2770, the boreal zone loses 150 PgC, while the tropical zone gains 52 PgC and the temperate zone 15 PgC (Fig. 10), resulting in a global loss in soil carbon. The significant loss in carbon in the boreal zone is mostly due to permafrost thaw and results in significant emissions into the atmosphere. In contrast, in the tropical and temperate zones, the decline in soil carbon pools due to higher temperatures and increased microbial activity is counterbalanced by augmented vegetation carbon, which in turn positively influences soil carbon levels through the conversion of vegetation to litter and subsequent incorporation into the soil carbon pool.

Also the contrast between short-term and long-term dynamics is most notable for boreal soil carbon pools (Figs. 9 and 10). For





instance, within the Upper CC/Upper LSC scenario, a short-term (1988-2100) soil carbon pool decrease by 19 PgC is observed (the tropics gain 26 PgC and the temperate zone gains 6.9 PgC). In the long-term (1988-2770), the boreal carbon losses are

strongly augmented with a decrease in the soil carbon pool by 110 PgC. The differences in the tropical zone (gain of 30 PgC) and the temperate zone (gain of 8.6 PgC) is relatively constant.

Due to the gradual decay of soil carbon subsequent to its peak after the transient phase, the long-term influence of land system change (LSC) on soil carbon is clearly recognizable, but with a relatively subdued effect (Figs. 9 and 10). At the Upper CC scenario soil carbon increases in the tropical zone between Lower LSC and Upper LSC from a gain of 21 PgC to a gain of 30

PgC in the tropical zone, from 6.8 to 8.6 PgC in the temperate zone and from $-130$ to $-110$ PgC in the boreal zone. This slight increase in soil carbon due to heightened LSC can be ascribed to the incorporation of decomposed biomass from deforestation into the soil carbon reservoir. Notably, owing to the gradual decay rate of soil carbon over an extended temporal span, the initial effects of land use change retain their influence on soil carbon levels over several hundred years. This phenomenon is particularly evident in colder temperate and boreal regions, where microbial activity exhibit a relatively slower pace compared

to the warmer tropical regions (Figs. 9 and 10).

### 3.5 Interplay of land system change and climate change: Cumulative effects and interactions

Climate change and land system change are two of the most important dimensions in the critical challenge of balancing the needs of human development and the preservation of natural systems. Within this context, it is important to recognize the inherent interconnection of both dimensions, as their influence on the Earth system frequently manifests as a cumulative or

net effect arising from their combined impact. For instance, LSC through deforestation and CC through the change of global radiative forcing can both impact regional and global temperature and precipitation in different ways: As forests play a vital role in the water cycle by releasing water vapor into the atmosphere through the process of transpiration (Bonan, 2008), the decrease in natural land led to a decline in water transpiration and consequently precipitation. For example, deforestation in the Amazon rainforest has been previously linked to reduced rainfall in the region (Marengo et al., 2009). This also led in turn

to the reduction of evaporative cooling, and therefore higher temperatures. This was specifically evident in the scenarios with transgressed LSC, where regions such as Brazil and Europe have a large expansion of managed land and show a substantial drying in their long-term climate equilibrium. On the other hand, cropland increases surface albedo compared to forests, resulting in a decrease in temperatures. Simultaneously, an increase of atmospheric $CO_2$ impacts the radiative balance of the planet and leads to a global increase in temperature. While in our experiments this effect is clearly dominant for global tem-

perature levels, it is also evident that the dynamics of local land-use change profoundly influence regional surface temperature dynamics. On the other hand, climate change's general ability to elevate precipitation, driven by the atmosphere's augmented moisture-holding capacity, is in contrast with the tendency of land use change to induce drier conditions.

Also, the carbon balance is affected by the accumulated effects of LSC and CC. Deforestation leads to a direct decline in vegetation carbon pools and a short-term increase in soil carbon and undermines the forests' capacity to sequester carbon. An

increased atmospheric $CO_2$ concentration and the connected climate impact alters vegetation productivity and mortality-related effects such as heat stress and wildfire occurrence (Flannigan et al., 2000; Boisvenue and Running, 2006), which in turn leads





to a change in vegetation distribution. Furthermore, the fertilization effect of increased atmospheric $CO_2$ and hence an increase of vegetation carbon stands in opposition to the sharp decline in vegetation carbon pools after large-scale deforestation in the transient phase. This change in biomass and vegetation structure in turn affects regional and global climate through biophysical

feedback mechanisms.

The net effect of our scenarios on carbon emissions to the atmosphere depends strongly on the time period after the impact: Deforestation has the strongest impact in the transient period and leads to a short-term source of land carbon and a leakage of $CO_2$ into the atmosphere, while the fertilization effect through elevated $CO_2$ is dominant in the following centuries of the experiment and results in a long-term increase of the land carbon sink. Therefore, the co-examination of both planetary

boundaries and the biophysical interactions between the atmosphere and biosphere are crucial, as the net effect of change often depends on counteracting influences.

### 3.6   Impact of land system change and climate change in different climatic zones

The transgression of planetary boundaries shows distinct effects across diverse climatic zones on Earth. Although CC exerts a

relatively consistent global influence, resulting in amplified temperatures and augmented precipitation, our investigation reveals noteworthy variations in impact in certain regions. The boreal zone, notably, exhibits the most profound increase in warming, attributed to the Arctic amplification phenomenon. The transgression of the CC planetary boundary in this region can lead to two distinct processes, potentially leading to a tipping point: The first pertains to a phenomenon termed "northern expansion", wherein trees rapidly extend their range towards the northern peripheries. This expansion has the effect of covering highly

reflective snow surfaces with tree canopies, a process that intensifies the rate of Arctic warming. The albedo effect comes into play here, where darker surfaces absorb more energy compared to lighter, reflective surfaces (McKay et al., 2022). The second process is referred to as "southern dieback". Here, boreal forests face destabilization over larger expanses, typically extending up to around 100 kilometers from their southern boundaries. This phenomenon results from a combination of factors, including alterations in hydrological patterns induced by warming, an increase in the frequency of fires, and outbreaks of bark

beetle infestations. These factors collectively contribute to the decline of boreal forests in these southern regions (McKay et al., 2022). We find that the boreal zone suffered from the most extensive soil carbon loss owing to permafrost thawing due to warming (Fig. 10).

In the tropical zone, the most substantial increase in vegetation carbon is observed, primarily due to the pronounced impact of the carbon fertilization effect. The carbon-rich rainforests, benefiting from ample water and light availability, emerge as

significant carbon sinks during the course of climate change. Despite heat-related detrimental impacts on biomass, the overall effect is insufficient to trigger widespread alterations in vegetation cover or a transition towards less vegetated regions (Drüke et al., 2021a). Furthermore, intriguing instances of greening in typically arid regions are identified, such as the Sahara desert, indicating potential shifts in the water cycle and a trajectory towards larger vegetation coverage (Fig. 7).

Comparatively, the impact of land use change (LSC) in our experiments is notably localized, predominantly observed in regions

where pristine forests are replaced by agricultural and pasture lands. This transformation is particularly evident in tropical and




temperate regions, for example in Europe and Brazil. The conversion of these areas into croplands or pastures results in a surface temperature increase that frequently surpasses the global average trend. Additionally, the decrease in transpiration leads to confined reductions in localized precipitation. Though these impacts tend to be regional, they hold the potential to trigger broader continental climatic shifts, influencing adjacent natural landscapes which can also impede potential forest recovery

by locking systems into warmer and drier conditions (Drüke et al., 2023). In contrast, the boreal zone exhibits a relatively minor susceptibility to the effects of land use change. This phenomenon primarily stems from the limited allocation of land for agricultural or cattle-rearing activities within this zone since its agro-ecological potential is relatively low.

## 3.7 The committed effect of LSC and CC in long-term climate projections

Although planetary boundaries are globally defined and are fixed and constant values, the repercussions of breaching these

thresholds have predominantly been studied for periods spanning the next few decades or have leaned towards expert elicitation (Chrysafi et al., 2022) and conceptual models (Anderies et al., 2013). Our study utilizes a fast yet comprehensive Earth system model, including detailed processes that link the biosphere and atmosphere, which allows us to study time periods of several hundred years within computationally reasonable time frames. This approach yields insights into the consequences of committed changes, as we simulate constant levels of planetary boundary transgression. In the initial decades, rapid change

of atmospheric $CO_2$ and deforestation result in temperature increments, and carbon losses (in scenarios with a large degree of LSC transgression). The long-term trajectory extends toward an even hotter planet, accompanied by an increase in vegetation carbon and the release of soil carbon into the atmosphere. While the changes in precipitation and vegetation carbon are most pronounced in the short-term period until 2100, soil carbon and temperatures show a centuries-long trend until almost the end of our study period. Even if humanity succeeds in stabilizing land system change and climate change subsequent to their initial

transgression (as in our scenarios), the impacts over the next centuries would lead to further warming and a steady leakage of carbon into the atmosphere. This discernible long-term influence, impacted by the ocean's capacity to buffer global temperatures and the gradual carbon dynamics within soil and vegetation, should not be neglected. Equally vital is the restoration and conservation of natural habitats, particularly forests, which assume a pivotal role in climate regulation and provide a multitude of benefits for both humanity and the natural world (Drüke et al., 2023; Nobre et al., 2016). A restoration scenario in our study

(Lower CC, Lower LSC), result in a large and long-term sink of atmospheric carbon in the biosphere and a stabilization of global temperature increase within acceptable boundaries.

## 4 Limitations of the modeling approach

In contrast to other studies on planetary boundaries that rely on conceptual frameworks, remote sensing data, or non-coupled models, our approach leverages the fully biophysically coupled Earth system model POEM, coupled to the advanced, dynamic

global vegetation model (DGVM) LPJmL. This modeling framework offers the opportunity for simulating long-term Earth system dynamics under a spectrum of climate change and land system alteration scenarios. While our model exhibits lower spatial resolution than the majority of CMIP6 models, it includes a sophisticated dynamic global vegetation model that cap-



tures the feedback loops between vegetation and climate, a feature absent in simpler modeling approaches (Lasslop et al., 2016; Baudena et al., 2010). Additionally, our model incorporates fire dynamics, permafrost dynamics, and an advanced land use scheme—attributes relatively scarce among CMIP6 models (IPCC, 2023). These features, combined with relatively low computation costs, allow us to probe the extended temporal evolution of carbon stocks and climate across varying levels of planetary boundaries related to land system transformations and climate shifts.

Nevertheless, a salient challenge in elucidating the extended dynamics of climate and carbon arises from the intrinsic uncertainty in representing land surface processes in Earth system models. While models like POEM are tuned and calibrated to align with historical data, projections into the future are characterized by considerable process uncertainty (Friedlingstein et al., 2014; IPCC, 2014). Biases in the simulation of global climate lead to a propagation of errors in the global vegetation distribution, cascading into effects on the carbon cycle, water cycle, and energy equilibrium. In simulations from the Coupled Model Intercomparison Project (CMIP) phases 5 and 6 (Friedlingstein et al., 2014; Eyring et al., 2016), differences in carbon fluxes between the biosphere and atmosphere, as well as surface temperatures, amount to approximately 800 PgC globally and 2°C in an idealized emission scenario over a century (Arora et al., 2020). Such discrepancies in global warming and vegetation carbon fluxes hold the potential to significantly impact outcomes for long-term carbon dynamics.

Efforts to mitigate the uncertainties of POEM and other Earth system models necessitate an enhanced comprehension of the mechanisms and feedback loops essential for climate-vegetation interactions. For instance, the well-documented phenomenon of carbon fertilization, which elevates photosynthesis rates and diminishes leaf transpiration in response to increased atmospheric carbon dioxide ($CO_2$), shows variability across plant species and ecosystems (Wang et al., 2020). Its manifestation depends on diverse factors encompassing water availability, nutrient supply, temperature, and plant adaptability. Nonetheless, the translation of carbon fertilization into heightened biomass productivity or carbon accumulation is non-uniform, given the existence of ecological constraints including tissue mortality, sapling viability, insect infestations, disease outbreaks, nutrient constraints, and climate extremes. Present-day vegetation models, including LPJmL, often fail to account for these intricacies. Consequently, it is imperative to deepen our comprehension of the interplay between $CO_2$ fertilization and ecological variables and to embed this understanding within more comprehensive and robust models, enabling more accurate projections regarding the future of terrestrial carbon sinks and their reciprocal influences on climate change.

Furthermore, considerable uncertainties and gaps persist in our comprehension of soil carbon responses to climate change (CC) and land system change (LSC), particularly on regional and local scales due to the lack of validation data and process understanding. Subsequent studies should enhance the representation of microbial processes, land use dynamics, and the feedback loop connecting soil carbon and climate within Earth system models. Moreover, the augmentation of soil carbon observations, characterized by improved quality and broader coverage across diverse biomes and land use categories, holds significant potential to advance our understanding of these complex dynamics.

While our present study focuses on the long-term response under a controlled and constant atmospheric $CO_2$ concentration, particularly in instances of substantial LSC transgression, our simulations indicate a considerable release of carbon from deforestation into the atmosphere. Future investigations hold the potential for incorporating the influence of carbon cycle interactions into the simulations. Such feedback mechanisms could entail emitted carbon elevating atmospheric $CO_2$ levels. This scenario





could lead to a potential cascade of positive feedback loops, thereby intensifying climate change. On the other hand, the significant carbon sinks present in oceans and terrestrial ecosystems possess the capacity to attenuate atmospheric $CO_2$ levels,

potentially countering the impact of anthropogenic emissions or even leading to reductions.

Another limitation of our study approach is the concentration solely on the impacts stemming from two distinct planetary boundaries — climate change and land system change. Other factors such as biodiversity loss, ocean acidification, and aerosol loading, although crucial, pose challenges in terms of interactive modeling. Future endeavors should be directed towards creating and advancing modeling capabilities for simulating a broader spectrum of planetary boundaries. By comprehensively

assessing their interconnected impacts over extended temporal scales, we would be able to better estimate the risks faced by our society when leaving from Earth system's safe operating space.

## 5   Conclusions

In this study, we employed a process-based Earth system model to investigate the consequences of different levels of transgression in the interacting planetary boundaries for climate change (CC) and land system change (LSC). We find that LSC

predominantly causes a pronounced decrease in short-term vegetation carbon and precipitation dynamics. CC, on the other hand, results mostly in a long-term temperature and vegetation carbon increase while it negatively impacts soil carbon pools. Intriguingly, the effects of CC and LSC often stand in opposition to one another; while climate change increases vegetation carbon through carbon fertilization, LSC (i.e. deforestation) decreases this pool. The combined net effect of CC and LSC varies across regions, with LSC resulting in the most significant impacts on tropical and temperate regions, whereas CC has its

strongest influence in the northern boreal zone. Here, Arctic amplification and the thawing of permafrost soils are found to be a key mechanism. Generally, at larger degrees of planetary boundary transgression, surface temperatures increase by 2.7-3.1 °C between 1988 and 2770 (under the transgressed CC boundary), and almost 150 PgC soil carbon is lost in the boreal zone during the same time period under transgressed CC and transgressed LSC.

Our findings underscore the importance of considering both short-term and long-term carbon dynamics when assessing the

repercussions of land use and climate change upon the Earth system. Moreover, our results emphasize the vital significance of concurrently modeling the combined impacts of CC and LSC, as this approach is pivotal for studying ecological interdependencies and atmosphere-land feedback mechanisms. By considering the complex interactions over extended temporal scales, we advance to a more comprehensive understanding of the true magnitude and risks associated with transgressing planetary boundaries.

It is extremely important to keep the planetary boundaries at least below the upper end of the zone of increasing risks. A further transgression could result in a catastrophic scenario for our planet, entailing a profound depletion of biodiversity, ecosystem services, human well-being, and societal equity. Therefore, an urgent and decisive response is needed to reverse the prevailing trends and reinstate the Earth's system to a safer condition. This transformation necessitates a comprehensive reevaluation of values, institutions, policies, and behaviors, as well as rapid and profound decarbonization across the economy and society

(Steffen et al., 2018).



*Code and data availability.*  The POEM code is available at https://doi.org/10.5281/zenodo.4700270. The output data will be made available upon request to the reviewers. For the final published paper, data and code will be made available via Zenodo.

*Author contributions.*  MD and WL conceived the study. MD performed and analyzed the simulations with support from WvB and SP. AT prepared the LSC input files. MD, WL, SS, and KT contributed to discussions and interpretation of the results with inputs from GF and SL.
MD wrote most of the paper with input from all co-authors.

*Competing interests.*  The authors declare that they have no conflict of interest.

*Acknowledgements.*  This work has been carried out as a contribution to the project POEM-PBSim: A Simulator for Earth's Planetary Boundaries, funded by the Volkswagen-Stiftung at the Potsdam Institute for Climate Impact Research. MD gratefully acknowledges the financial support of the Volkswagenstiftung. The authors gratefully acknowledge the European Regional Development Fund (ERDF), the German
Federal Ministry of Education and Research, and the Land Brandenburg for supporting this project by providing resources on the high-performance computer system at the Potsdam Institute for Climate Impact Research. The authors are grateful to the whole POEM development team at the Potsdam Institute for Climate Impact Research and to Johan Rockström for his leadership on planetary boundary science.



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
