# Peer review of "The long-term impact of transgressing planetary boundaries on biophysical atmosphere-land interactions"

_EGUsphere, 2023_

## Author Comment (AC1)

**Response Letter for "The long-term impact of transgressing planetary boundaries on biophysical atmosphere-land interactions"**

**Reviewer 1**

The research conducted by Drüke et al. offers insights into the Earth's future using the fully coupled and dynamic Earth system model, POEM. This study underscores the importance of respecting Earth's 'boundaries', demonstrating the extensive consequences of human-induced land use and climate change on the biosphere if these boundaries are overlooked or breached at various degrees. What sets this study apart is its scope: while the IPCC offers emission scenarios on climate change up to 2100, Drüke et al. delve into more extended time frames, revealing the potential long-term repercussions of human activities on both the Earth's biosphere and climate. Through detailed, spatially-explicit maps, the authors highlight both global patterns and regional disparities, showing the extensive ways human activities can alter land surfaces and climates.This work has substantial implications on the different levels of commitment that can serve as a guidance on safeguarding the planetary boundaries of land surface and climate change from a scientific point of view.

We thank the Reviewer for the positive comments about our study.

 However, I'd like to address two main concerns:

 First, the prescribed set-ups of climate change (CC) and land system change (LSC) are separated in the six different scenarios. The implicit assumption behind this set-up is that these two processes are relatively independent. For example, a scenario where there is low human activity but extensive climate change is presented by pairing low LSC with high CC. However, climate change and land use are interconnected: land use change influence global greenhouse gas levels, thereby affecting climate change. Conversely, a shifting climate also impacts agriculture and land use patterns. The intimate relationship between CC and LSC needs to be factored into the model. My concern is whether all the scenarios are realistic? And how does the model account for the interaction between CC and LSC? Secondly, as showcased in Figure 2c, when LSC boundaries are crossed, vegetation carbon storage experiences a significant reduction of approximately 180 PgC due to intensive land use. Soil carbon sees only a modest increase of 60-70 PgC, attributed to an increase in litter input into the soil carbon pool, as the authors have suggested. This leaves around 110 PgC of carbon to be either burned, utilized, or decomposed, inevitably increasing emissions. Even if only half of this carbon is released into the atmosphere, it would amount to a roughly 115 ppm (55 PgC) surge in atmospheric CO2—a significant increment given the maximum CO2 level in the most drastic scenarios is 550 ppm in the modeling setup. The study's modeling efforts lock the

CO2 level at a constant from 2052 onward, but for a comprehensive understanding of long-term changes, this factor should be integrated.

The chosen scenario framework closely resembles that of the IPCC's RCP scenarios, which prescribe a $CO_2$ trajectory and analyze the carbon cycle's response separately. However, instead of examining trajectories of $CO_2$ concentrations resulting from unspecified, compatible $CO_2$ emission trajectories, we set atmospheric $CO_2$ concentrations to values defined by planetary boundaries or fixed magnitudes of transgression. While RCP trajectories are linked to the underlying SSP scenarios, we here chose a different, more systematic approach to stress-test the planetary boundaries framework. We therefore chose to exert pressure on the climate and land-use boundaries that correspond with crucial inflection points of their definition (lower boundary, upper boundary, transgression). The specific objective is to investigate the long-term implications of such fixed forcings, focusing on the chosen values for each of the two studied planetary boundaries. This approach can be likened to sending a predefined signal into a complex electronic device with unknown properties to study resulting signals in different parts of the device, enabling conclusions about the device's properties.

Our investigation serves a distinct system analytical purpose, elucidating how Earth system components react to a long-term constant forcing of a defined magnitude. Similar to the IPCC's analysis of carbon cycle responses to RCP forcing, our simulations allow vegetation dynamics to adapt freely, considering various compartments and spatial distribution adjustments in response to this constant forcing. These adaptations occur through feedback mechanisms from the evolving climate system, excluding feedbacks that would modify the studied forcing's magnitude ($CO_2$).

Examples for included feedback mechanisms are for example the impact of different prescribed land use patterns on the energy and water balance through changes in albedo, evapotranspiration and roughness lengths.

While conducting experiments with a freely evolving setup, deviating from initial forcing values, could be valuable, such experiments would serve a different purpose. Such an experiment would be very different from design and yield very different results. While it is true that a huge amount of $CO_2$ released from the biosphere would increase emissions, over the course of our simulation setup the ocean sink could be more than adequate to compensate for these emissions and would most probably lead to a decrease in atmospheric $CO_2$ without further sources of carbon. To utilize a more realistic scenario, we would need to prescribe anthropogenic emission scenarios, which would add to the natural ones. This would involve a multitude of socioeconomic assumptions and analyses to be put into scenario setup, which was not the aim of this study. Furthermore, emission scenarios over the course of almost

1000 years are extremely unreliable and would provide a false realism from which we could learn far less than by using the approach of this study.

We have added some text to the introduction to clarify this (l. 81-84):

*"Unlike exploring unspecified emission trajectories, we fix $CO_2$ concentrations based on planetary boundaries. Our goal is to examine the lasting implications, emphasizing specific values for two planetary boundaries. While this approach does not follow a realistic future trajectory of the Earth system, it is comparable to studying a predefined signal in a complex electronic device to understand its properties. Our research sheds light on Earth system responses to a constant, defined forcing."*

Lastly, a few specific comments on the manuscript:

1. The manuscript could benefit from an early and clear definition of 'transgressing', as it's a central theme.

We thank the reviewer for this useful comment and adapted the description of the two simulated planetary boundaries to focus more on transgression (l. 39-53)

*"Land system change, which includes deforestation and urbanization, is a key process that has been defined as one of the nine planetary boundaries. It is a major driver of environmental change by impacting the functioning of ecosystems and contributing to the loss of biodiversity (Díaz et al., 2019).*

*The planetary boundary for land system change is defined through the observable proxy given by the extent of remaining forest cover (50%, 85%, and 85%) of the three major forest biomes – temperate, tropical, and boreal, respectively (Steffen et al., 2015). Transgression of this boundary means going beyond the upper end of the zone of increasing risk of land use and forest cover (20%, 40% and 40% remaining forest extent for temperate, tropical and boreal, respectively), which can have a range of other impacts, such as altering the carbon cycle and contributing to climate change (IPCC, 2023), as well as links to the hydrological cycle and moisture recycling, biogeochemical flows and aerosol emissions, which are beyond the scope of this study.*

*Climate change can disrupt the balance of ecosystems and result in shifting global temperature and precipitation patterns. It is also expected to lead to increasing frequency and severity of extreme events and altered availability of water resources (IPCC, 2023). The planetary boundary framework sets the lower boundary of the uncertainty range for the transgression of atmospheric CO2 concentration at 350 ppm, a value that was reached in 1988 (Rockström et al., 2009; Steffen et al., 2015).*

*Transgression of the climate change boundary means surpassing the upper boundary of increasing risk in atmospheric CO2 concentration at 450 ppm, which can trigger dangerous feedbacks and tipping points in the Earth system (Rockström et al., 2009; Steffen et al., 2015). "*

2. Lines 88-90: Consider adding a sentence clarifying that the previous static vegetation model component has been superseded by LPJmL5, which permits dynamic vegetation changes.

We added the following (l. 97-100):

*"While AM2 and MOM5 remain dynamically coupled to the model, the simple land model LM/LM2 with static vegetation has been replaced by the more advanced global vegetation model LPJmL5, which includes dynamic vegetation and advanced land surface processes."*

3. Line 130: Would be better to give a brief explanation on why aerosols etc. remain at 2003 level.

Model-specific forcing datasets for CM2Mc were not available to the date of conducting these experiments. As these forcings are spatially explicit and comprise a multitude of e.g. different Aerosol species, we were not able to create the fitting data sets. Our focus was on greenhouse gas forcing, which is the dominant driver of atmospheric forcing in the recent decades. To clarify, we added to the text (l. 144-146):

*"Due to data availability, from 2004 on, only greenhouse gas forcing is changed, while minor drivers of changes in atmospheric forcing (aerosols, solar radiation, and ozone) are set to their corresponding 2003 values."*

4. Line 140: I'd recommend introducing the definition of 'transgressing' in the beginning for the readers to grasp this pivotal term from the get-go.

see 1.)

---

## Author Comment (AC2)

**Response Letter for "The long-term impact of transgressing planetary boundaries on biophysical atmosphere-land interactions"**

**Reviewer 2**

The manuscript addresses some of the criticisms of the Planetary Boundary framework, especially the oversimplification of a very complex topic and the missing interactions between boundaries. Given the prevalence of the PB framework in the public debate on climate change, it provides a very valuable contribution to the discussion. I have some major concerns that should be addressed before publication.

We thank the Reviewer for the useful comments and review of our study.

My main concerns are with the validation of the model results:

- The model used in the study tries to model highly chaotic processes over very long periods of time, especially deforestation (which depends on human behavior) and vegetation itself. The authors use single model runs, a sensitivity analysis is necessary to show uncertainty of the model results. Especially as running the model does not seem to be too expensive this should not be too difficult.

Although our model is fast in comparison to other GCMs, conducting entire sensitivity studies is still unfeasible. Despite utilizing a powerful high-performance computing system, a single 800-year run demands at least four weeks, excluding the prolonged spin-up of carbon pools, which scenarios can utilize collectively. This limitation is characteristic of sophisticated Earth system models. The predicament aligns with established practices, such as the IPCC's approach of evaluating multiple models under a common protocol. Although an intercomparison would be valuable, proposing it for the 7th IPCC assessment report is more suitable, given the comprehensive nature of such an endeavor. However, this scope surpasses the intentions of an application paper on simulating planetary boundaries, where the scenario simulations serve a partially analytical purpose. A comprehensive analysis warrants a separate and dedicated paper.

We added a sentence to the computational demand of the model to the methods (l. 184-189):

*"Utilizing a robust yet relatively efficient Earth system model allows us to conduct simulations spanning centuries, a departure from typical CMIP6 experiments that generally conclude in 2100. Nevertheless, even with CM2Mc-LPJmL, a single 800-year run requires a minimum of four weeks (utilizing 64 CPUs), excluding the extended spin-up period for carbon pools, which scenarios can collectively share. Consequently,*

*our emphasis is on no more than six distinct experiments, aiming to scrutinize the influence of various combinations of planetary boundaries on the Earth system. This approach already poses a considerable demand on computational resources."*

- The model is quite novel. How well does hindcasting work? This is probably the best way to verify the accuracy of model results.

The validation and evaluation of the model has been done in the model description paper (Drüke et al. 2021). We added the following figures, adapted from Drüke et al. 2021 to the Supplement. They show an evaluation of the historic period. While there are some deviations, mostly due to the coarse spatial resolution (e.g. in the exact location of the ITZC), modeled biomass and historic temperature evolution are relatively well captured for an Earth system model with coarse resolution and dynamic vegetation. For more details refer to Drüke et al. 2021.

We added in l. 119-121 :

*"POEM exhibits significant climate biases due to its coarse resolution (as shown in Fig. S5). However, it does reasonably well in capturing historical temperature trends (Fig. S6) and global biomass distribution (Fig. S7), albeit with a pronounced negative bias in Amazon rainforest biomass. For a more comprehensive evaluation of the model, refer to Drüke et al. 2021."*

[Figure]

**Fig. S5:** (a) Global surface temperature from CM2Mc-LPJmL averaged over the period 1994–2003, (b) global precipitation from CM2Mc-LPJmL averaged over the period 1994–2003, (c) zonal mean temperature from CM2Mc-LPJmL (red line) and ERA5 data (blue line) averaged over the period 1994–2003, (d) zonal mean temperature from CM2Mc-LPJmL (red line) and ERA5 data (blue line) averaged over the period 1994–2003. Plot is analogous to Drüke et al. 2021.

[Figure]

**Fig. S6:** Yearly and decadal global mean temperature anomaly (relative to the reference period 1951–1980) of CM2Mc-LPJmL compared to GISTEMP data from 1880–2018. Note that, from 2004 on, only greenhouse gas forcing remains, while aerosols, solar radiation and ozone are set to their corresponding 2003 values. Plot is analogous to Drüke et al. 2021.

[Figure]

**Fig. S7:** (a) Mean global above-ground biomass of GlobBiomass (Santoro et al., 2020) evaluation data. (b) Mean global above-ground biomass of CM2Mc-LPJmL over the period 2006-2015. (c) Difference of the above-ground biomass between CM2Mc-LPJmL and GlobBiomass evaluation data. Blue/red colors denote an overestimation/underestimation of biomass by CM2Mc-LPJmL. (d) Latitudinal sum of above-ground biomass from CM2Mc-LPJmL (blue line, R2=0.65, NME=0.58) and GlobBiomass evaluation data (red line). Plot is analogous to Drüke et al. 2021.

- I think it is important to see the isolated effect of crossing boundaries (and I disagree with reviewer 1 here), especially because the authors put interactions between the different boundaries as one of the main aspects of your analysis. It would have been interesting to see "risky land system change" and "safe" climate change scenario, even though this is an impossible scenario. This is the only way to quantify the additional effects of crossing both boundaries.

We agree with the reviewer that simulating the full matrix of different boundary states would be interesting but as outlined earlier, our experiments actually used a lot of computational resources. As our main focus was on the analysis of long term dynamics, we decided to focus on the more probable states of the planetary boundaries and following a prioritization of computational resources.  Unfortunately,

doing these experiments would take up too much computational power and was therefore out of scope for this study.

I have one minor remark: In some figure the font is unreadably small

We adapted the figures and increased the font size in figures 4,6,8 and 10.

---

## Author Response (AR2)

Dear Prof. Christian Franzke,

Thank you for your response and for your positive feedback regarding the revised manuscript. We appreciate the thorough review process and are glad to hear that the reviewer now recommends accepting the manuscript for publication.
Regarding the clarification of the two sentences you mentioned:

Line 45: We apologize for any confusion caused by the wording. The phrase "zone of increasing risk" means that the Earth system is at a high risk of exiting the safe operating space and entering an unstable state. The exact threshold of exiting the stable state that resembles the Holocene epoch is uncertain, so we use zones of lower and higher risks based on the planetary boundary framework.
We appreciate your feedback and have revised the sentence accordingly:

*"Transgression of this boundary (20\%, 40\% and 40\% remaining forest extent for temperate, tropical and boreal, respectively),* **means going beyond the upper end of the zone of increasing risk to the stability of the Earth system,** *which can have a range of other impacts, such as altering the carbon cycle and contributing to climate change (IPCC, 2023), as well as links to the hydrological cycle and moisture recycling, biogeochemical flows and aerosol emissions, which are beyond the scope of this study."*

Line 155 we changed to:

*"In the scenario names, "Lower" and "Upper"* **refer to the zone of increasing risk to the stability of the Earth system***, with the "Lower" end representing the boundary defining the "safe operating space" and "Upper" representing the transition from risky to dangerous."*

Thank you again for your valuable feedback.

Best regards,

Markus Drüke
on behalf of the Co-Authors